# Integrating Absolute Sustainability and Social Sustainability in the Digital Product Passport to Promote Industry 5.0

Luigi Panza [ID], Giulia Bruno *[ID] and Franco Lombardi [ID]

Department of Management and Production Engineering, Politecnico di Torino, 10129 Turin, Italy; luigi.panza@polito.it (L.P.); franco.lombardi@polito.it (F.L.)
* Correspondence: giulia.bruno@polito.it

**Abstract:** The establishment of the digital product passport is regarded to be a prominent tool to promote environmental and social sustainability, thus enabling the transition towards Industry 5.0. In this way, it represents a holistic tool for the decision-making process of several actors of a product's value chain. However, its development is still ongoing and the absolute perspective of environmental sustainability and the social sustainability have been overlooked. The present work aims to fill these gaps and complement the literature currently available on the digital product passport with a threefold purpose. Firstly, by referring to social life cycle assessment methodologies, useful social indicators to include in the digital product passport are discussed and proposed. Secondly, the need for an absolute perspective of environmental sustainability that respects the natural limits of our planet is presented; based on the LCA methodology and the Planetary Boundaries framework, environmental attributes and environmental impact indicators with the corresponding threshold are proposed to be included in the passport and enable the so-called absolute environmental sustainability assessment of products. Finally, a framework based on a cyber-physical system for filling in the digital product passport throughout a product lifecycle is conceived. This work represents an example of how the hallmark technologies of Industry 4.0 can be used towards Industry 5.0.

**Keywords:** Industry 4.0; Industry 5.0; digital product passport; absolute sustainability; social sustainability; circular economy; cyber physical systems

## 1. Introduction

Human activities are causing several ecological crises, mainly due to the unsustainable use of the planet's resources. The adoption of an absolute perspective of environmental sustainability in which economies and societies develop within the Earth's carrying capacity is a mandatory condition for ensuring sustainable development [1]. According to Bjørn and Hauschild [2], the carrying capacity can be defined as 'the maximum sustained environmental intervention a natural system can withstand without experiencing negative changes in structure or functioning that are difficult or impossible to revert'. The carrying capacity of our Earth can be represented by nine life support systems, in which each of them is controlled by specific quantitative variables. The threshold values of such variables represent the Planetary Boundaries (PBs) that define a 'safe operating space' for the human development [3]. Such a novel line of thinking—known as absolute environmental sustainability and envisioned by Hauschild et al. in 2017 [4]—entails the development of products that achieve environmental sustainability in absolute terms without exceeding the 'safe operating space', by respecting the natural limit set by our planet.

However, in 2022, five out of nine of the planetary boundaries have already been transgressed [5].

Between 2008 and 2018, the industry sector played a pivotal role in the European Union's (EU) economy by representing 20% of the Gross Domestic Product (GDP). This made it the largest single contributor to the EU's economic growth during that period [6].

According to Eurostat [7], in the second quarter of 2022, the manufacturing sector accounted for 23% of the EU's carbon footprint. This evidence highlights how the implementation of the well-known Industry 4.0 paradigm has been mainly focused on increasing efficiency and flexibility in manufacturing through the introduction of digitalization and connectivity, without a primary focus on sustainability [8]. In order to achieve the targets on climate change by 2030, the European Commission (EC) foresaw that industry should reduce its carbon footprint by between 18.2 and 25.1% [9].

To meet this challenge, the need to shift from Industry 4.0 to Industry 5.0 is increasingly being acknowledged. The concept of Industry 5.0 does not rely on a technological revolution, as Industry 4.0 did, but on a values revolution, in which the use of the typical technologies of the fourth industrial revolution has a broader purpose than just maximizing efficiency and enterprise profit [6]. This wider scope can be represented by three key elements: human-centricity, sustainability, and resilience [10], as schematically depicted in Figure 1.

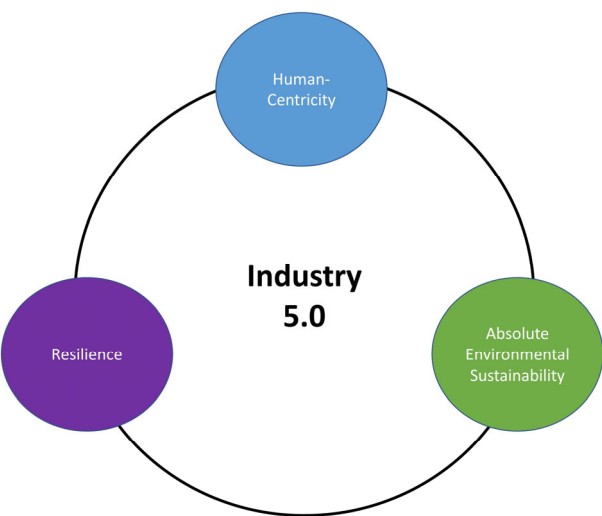

**Figure 1.** Concept of Industry 5.0, rearranged from [10].

Human-centricity entails putting the needs and interests of all the stakeholders interacting with the industry at the core of the industrial purpose. The needs of customers, workers, value chain's actors, local communities, and society need to be properly identified and addressed in each phase of a technology lifecycle in order to promote the well-being of people [11].

Absolute environmental sustainability, as described earlier, aims to develop products and technologies that operate within the natural limits of our planet, represented by the PBs. The term 'Absolute' has been added in Figure 1 to emphasize the need of an absolute perspective of environmental sustainability.

Resilience refers to the capacity of industry to face unexpected challenges and disruptions [10]. For instance, energy transition can potentially replace the dependence on fossil fuels with a new type of dependence based on raw materials. Indeed, most green technologies, such as batteries, wind turbines, and solar photovoltaic, to name just a few, rely on so-called Critical Raw Materials (CRMs), which are resources with a high risk of supply disruption for the EU economy [12].

It turns out that both absolute environmental sustainability and resilience can be supported by the implementation of Circular Economy (CE) business models, which entail re-introducing in the value chain products, components, or materials when they reach their end of life. In this way, the pressure on the PBs can be reduced, as well as the risk of supply chain disruption [12,13]. However, an effective implementation of a CE requires an efficient information flow between several actors of the value chain in order to track the location, availability, and conditions of the resources belonging to a product [14]. Indeed, the lack of

information along a product's value chain is regarded as one of the main factors hindering the adoption of circular strategies [15].

Industry 5.0 can address this challenge by exploiting the digital technologies typical of the fourth industrial revolution. The creation of a digital identity of materials, namely the digital material passport, would allow all of the relevant material information throughout the material lifecycle to be recorded and tracked, from the sourcing phase to the end of use [13]. In such a way, the information related to the environmental footprint of a material lifecycle, as well as its conditions, would be properly stored and made available for the implementation of CE strategies. The digital MP can be regarded as an essential part of a broader digital information system—that is, the Digital Product Passport (DPP) envisioned by the EC [16,17]. The aim of the DPP is to foster sustainability, enable the transition to a CE, and support consumers in making responsible purchasing choices through the product-related information recorded along its lifecycle.

The development of the DPP is ongoing and the first works on this topic are beginning to appear in the literature. Specifically, the first research efforts have been oriented on studying the information that will be included in the DPP and the development of conceptual frameworks for data collection and management. However, as further detailed in the next section, no work to date has addressed the issue of including an absolute perspective of environmental sustainability in the DPP; in addition, there is a scarcity of studies addressing social sustainability issues, and even when they do, they tend to provide only general information.

This work aims to address these gaps and answer the following research questions (RQs):

**RQ1.** *What social information will be stored in the DPP?*

**RQ2.** *What absolute environmental information will be stored in the DPP?*

Concerning the first RQ, there is, at present, no established standard methodology for conducting a social life cycle assessment of products. The current work primarily relies on recommendations from the United Nations Environmental Program (UNEP) [11] and utilizes the Product Social Impact Life Cycle Assessment (PSILCA) methodology [18], that is also endorsed by the UNEP. These methodologies are employed to extract the relevant social indicators to be incorporated into the DPP, as discussed in Section 4.

Relating to the second RQ, in contrast to the social life cycle assessment, the environmental life cycle assessment of products can rely on a well-established methodology, commonly known as the Life Cycle Assessment (LCA), standardized by ISO 14044:2006 [19]. The LCA methodology allows the environmental impacts throughout a product's lifecycle to be systematically categorized and quantified. However, the traditional LCA methodology does not provide a means to identify the allowable limits for these environmental impacts, making it impossible to assess whether a product is sustainable in absolute terms [20]. The existing literature only shows initial efforts in this regard, primarily focusing on the integration of the PBs framework into the LCA methodology. Section 5 presents the environmental and technical data to be integrated into the DPP, building on the above-mentioned methodologies. These data serve to complement the initial research efforts conducted by the authors [13].

Finally, the current work also aims to support the digitalization of the product lifecycle to promote the effective implementation of the DPP. Thus, this study will address a third RQ:

**RQ3.** *How can the product lifecycle be digitalized to collect useful data for the DPP?*

To answer this question, a framework based on Cyber-Physical Systems (CPSs) is proposed to enable the automated collection of technical and environmental data into the DPP. The framework is built on the 5C architecture, which is widely recognized and extensively employed in the existing literature [21]. The conceived framework also serves as a noteworthy illustration of how a fundamental pillar of Industry 4.0 can be leveraged to facilitate the transition to Industry 5.0.

The article is structured as follows. Section 2 presents the available works in the literature concerning the DPP. Section 3 provides a general background on the current state of the DPP in terms of the expected advantages, essential requirements, technical implementation, and general information to be stored. Section 4 describes the social information to be stored in the DPP to promote the human-centricity feature of Industry 5.0. Section 5 presents the information to be stored in the DPP to promote absolute environmental sustainability and a circular economy, while Section 6 describes the framework for automatically collecting the environmental and technical data throughout the product lifecycle. Finally, Section 7 concludes the article and proposes possible future research directions.

## 2. Related Works

The works available in the literature mainly study the types of data and information to be included in the DPP and different frameworks for their management.

The literature was analyzed by consulting the articles available in the Google Scholar database concerning the keyword 'Digital Product Passport' and by referring only to journal articles and conference papers in the English language. All of the papers have been read, and the most relevant ones have been included in the present section and summarized in the following.

King et al. conceptualized the digital product passport as a collaborative and socio-technical system of systems. This innovative approach involves shared ownership among various stakeholders throughout the product lifecycle. They analyzed the requirements relating to this new kind of system-based concept [22]. Plociennik et al. proposed a framework for a digital lifecycle passport that relies on the asset administration shell and a cloud-based application. They also implemented a case study to demonstrate the improvement of the sorting process of electronic waste [23]. Adirson et al., through desk research and stakeholder workshops, discussed some design issues of the DPP with the expected benefits for the stakeholders of a product value chain [24]. Koppelaar et al. presented a conceptual design of the DPP for circular supply chain management. Their work aimed to facilitate the recovery and reuse of CRMs at both the component level and material levels. They conducted an assessment of the existing information management practices for critical raw materials and surveyed the information needs of 10 manufacturers, producer responsibility organizations, collectors, and recyclers. Based on these needs, 14 key processes and exchanges for product information management were identified, forming the basis of the Digital Product Passport circular supply management system [25].

Berger and colleagues presented a conceptual framework aimed at creating a digital battery passport for electric vehicles. Their work focused on providing the necessary information to effectively implement the digital battery passport, considering four distinct perspectives, which include: (i) battery; (ii) sustainability and circularity; (iii) diagnostics, performance, and maintenance; (iv) value chain actors. The objective was to facilitate the decision-making process for the sustainable management of electric batteries [26]. The same authors presented a conceptual framework that showcases how data science and machine learning techniques can facilitate the secure exchange of data among stakeholders in the electric vehicle battery (EVB) value chain in the context of the DPP. By enabling confidentiality-preserving data exchange, their framework aimed to address concerns related to data sharing and to enhance sustainability data management for EVBs [27].

A classification system for product data in the DPP was introduced by Stratmann et al. The development of this system involved a systematic literature review and a case study conducted on typical operational information systems. The classification system comprises three levels and encompasses 62 data points, which are organized into four main categories: (1) Product information; (2) Utilization information; (3) Value chain information; (4) Sustainability information [28].

Jensen et al. investigated the data requirements for digital product passports in a mechatronics context. They identified seven clusters of data: (1) usage and maintenance; (2) product identification; (3) products and materials; (4) guidelines and manuals;

(5) supply chain and reverse logistics; (6) environmental data; (7) compliance. A survey was conducted to evaluate the significance, accessibility, and sensitivity of specific data points, involving three original equipment manufacturers (OEMs), their customers, service partners, suppliers, and third-party recycling companies. The findings highlighted distinct data needs across these stakeholders, but indicated that the exchange of data and the supporting infrastructure for closing resource loops are still at a relatively early stage of development [29].

Jansen et al. studied the requirements for a DPP system based on stakeholder involvement and the literature from science and industry. They identified eight different groups of requirements: (1) legal obligations; (2) functional suitability; (3) security, confidentiality, and IP protection; (4) interoperability; (5) modularity and modifiability; (6) accessibility; (7) availability and time behavior; (8) portability. For each of this group, a list of requirements is provided [30].

Table 1 summarizes the current literature available in the context of the DPP.

**Table 1.** Summary of the current literature regarding the DPP.

| Research Work | Main Contribution |
| --- | --- |
| King et al. [22] | Definition of Digital Product Passport Ecosystem (DPPE), identification of nine system capabilities and corresponding information requirements. |
| Plociennik et al. [23] | Proposal of a Digital Lifecycle Passport (DLCP) managed by a cloud-based app and readable for both human and machines. Proposal of a use case in the sorting process of electronic wastes. |
| Adirson et al. [24] | Execution of desk research and stakeholder workshop for the identification of different DPP design options and identification of key issues to further investigate. |
| Koppelaar et al. [25] | Conceptual design of DPP focused on the recovery of critical raw materials (CRMs). Proposal of IT architecture for circular supply chain management. |
| Berger et al. [26] | Conceptualization of a Digital Battery Passport (DBP) and definition of the information requirements. Presentation of four different use cases to illustrate the benefits of the DBP for the value chain stakeholders. |
| Berger et al. [27] | Use of Data Science and Machine Learning approaches to support data exchange among electric vehicle battery stakeholders towards the DBP. |
| Stratmann et al. [28] | Classification of product data for the DPP in the manufacturing industry based on a systematic literature review and case-study research. Development of a use case in the machinery sector to show the applicability of the classification approach. |
| Jensen et al. [29] | Definition of data requirements for the DPP in the mechatronic industry based on interviews with different value chain stakeholders. |
| Jansen et al. [30] | Definition of the DPP requirements based on stakeholders' involvement and literature from science and industry. Identification of two important gaps: energy and resource utilization of DPP system and data privacy. |

All of these works represent valuable contributions to the development of the DPP. Different information requirements have been conceptualized and some use cases have been presented in the literature. However, both the absolute perspective of environmental sustainability and the aspect of social sustainability have not been addressed. This study aims to bridge these existing gaps. A set of social indicators that can be incorporated into the DPP framework is proposed, facilitating a thorough social assessment of product lifecycles. Furthermore, in addition to integrating social indicators, the inclusion of environmental impact indicators is proposed, along with the associated threshold values derived from the PBs framework, to enable an absolute assessment of product environmental sustainability.

## 3. Current State of the Digital Product Passport

In order to provide a general background on the current state of the DPP, the expected advantages, the essential requirements, the technical implementation, and the general information structure of the DPP are described in the following.

### 3.1. Expected Advantages

According to the works [16,17,23,31], the expected advantages from the creation of the DPP are multiple. Designers and engineers can improve the eco-design properties of products—such as the durability, reliability, reusability, repairability, re-manufacturability, recyclability, energy use, and generated waste—based on the life cycle information stored in the DPP. Manufacturers can establish traceability for warranty claims and recalls, repairers and maintenance services can provide better services thanks to the technical and historical information stored in the DPP, remanufacturers will benefit from the access to information related to disassembly and the health state of components, and recyclers will take advantage of the information on hazardous or valuable materials. The end-users will also benefit from the existence of a DPP because they can make more responsible choices during the purchase process thanks to the visualization of the product's environmental performance.

Moreover, the DPP can serve as a valuable compliance tool. In recent years, businesses operating within the European Union and the United States have been mandated to publish Sustainability Reports (SRs) detailing their activities. The information contained in the DPP can serve as evidence of compliance with legal requirements. This information can be incorporated into the SR, enhancing company's brand image and corporate reputation. Additionally, third-party organizations can utilize the DPP information to evaluate sustainability indices. Demonstrating a strong performance in this domain can translate into a competitive advantage [13].

### 3.2. Essential Requirements

The EC established the general requirements for the creation of the DPP. The DPP should be connected to a unique product identifier through a data carrier, which should be placed physically on the product; the data carrier and the unique product identifier should comply with standard ('ISO/IEC') 15459:2015 [32].

The data contained within the DPP must adhere to specific guidelines. It should be built on open standards, designed in an interoperable format, and be machine-readable, structured, and searchable. For a comprehensive overview of the general requirements, please refer to [17].

The EC also establishes the necessary technical requirements. The DPP is expected to be interoperable with other DPPs, ensuring seamless end-to-end communication and data transfer across technical, semantic, and organizational aspects. The users of the DPP will have free access to the information contained in the DPP based on their respective access rights. Data authentication, reliability, and integrity will be ensured, as well as security and privacy, and fraud will be avoided [17]. Ospital et al. emphasized the significance of establishing product traceability to ensure transparency within the DPP and to achieve the sustainability goals [33]. Guth-Orlowski highlights the need for inclusivity and flexibility of the DPP architecture. Participating in the DPP should not incur excessively high costs or present technical challenges that would exclude small economic actors. A flexible structure of the DPP is needed to face the dynamic of the current global value chains. This means that new actors or product attributes will be easily added, and actors and information will no longer be easily removed [31].

The aforementioned requirements have also been proposed and extended in the recent study by Boukhatmi et al. [34], wherein they identified several requirements based on practical insights from interviews and surveys. The requirements were grouped into 11 categories, as follows: (i) accessibility, i.e., grant complete platform access exclusively to specific parties via login process; (ii) completeness, i.e., provide additional information to the already present technical datasheets; (iii) consistency, i.e., promote uniform data

sharing among stakeholders involved in the value chain; (iv) efficiency, i.e., streamline operational processes for the collection of product information; (v) inter-operability, i.e., allow the smooth integration of new information from different stakeholders; (vi) security, i.e., guarantee the hosting of the platform by an external and impartial entity; (vii) sensitivity, i.e., restrict access to sensitive information; (viii) traceability, i.e., provide the history of the product life cycle via comprehensive data tracking; (ix) transparency, i.e., establish openness through means of sharing data; (x) time performance, i.e., reduce the time and effort spent by stakeholders in collecting information about the product; (xi) visibility, i.e., set up distinct levels of information visibility according to specific stakeholder groups.

To fulfil the aforementioned requirements, the following principles can be followed [30]:

- Data responsibility of the originators: The involved stakeholders of the value chain make statements about the activity they perform on the product. The statements are structured in a data sheet.
- Collection of data sheets: the DPP basically consists of a collection of data sheets produced by the stakeholders of the value chain, from material production to the end of product usage.
- Decentralized storage: The storage location is determined by the stakeholders themselves. In this way, the DPP is stored in distributed data stores. There is no central system that manages the DPP information.
- Verifiability of the data: The stakeholder's digital signature makes the data sheets verifiable. In this way, it is possible to identify who issued the information provided about the product and it cannot be denied. This enhances the level of trust in the DPP and in the data quality. A signed data sheet constitutes a certificate.

Finally, it is worth noting that the effective implementation of the DPP also requires suitable governance to be in place [35]. This includes abundant regulations, the EC taking an active role as a platform or registry manager, along with vigilant oversight from the relevant authorities [36].

### 3.3. Technical Implementation of Digital Product Passport

Guth-Orlowski proposes the use of decentralized technologies for the digital implementation of the product passport. Digital Decentralized Identifiers (DID) are applicable for humans, products, machines, processes, or companies and are usually anchored on a blockchain. In such a case, software for the creation and management of decentralized identifiers is needed and all the stakeholders of the value chain can create their own identifiers. At present, DID are standardized by the World Wide Web Consortium (W3C), which is one of the most important standardization bodies for Internet protocols.

Verifiable Credentials (VC) are machine-readable certificates that are signed with the private key of a DID. VC can be verified by signature; therefore, the data integrity is ensured, as well as the access control. Indeed, each stakeholder has the possibility to control access to certificates so that any company secret is protected from competitors and to allow the access only to an auditor, customer, or ministry. Because of this, VC can be employed to implement the certificates for the DPP. As stated before, a wallet can generate and manage DID, but it can be also used to create, sign, and manage VC. The referencing of one VC to another is called "credential chaining" and it represents the link of the DPP information.

The broad implementation of the DPP overseen by the EC requires that each company in the value chain has a wallet. The communication between several wallets is based on standards, so this ensures that vendor lock-in is avoided. Thanks to the wallet, each company can create its DID, collect VC about its activities, and it can also issue VC about other enterprises and products, for example, concerning the disassembly mode, repairing instructions, and so forth.

The use of a decentralized technology can give several advantages, such as end-to-end verifiability, data quality, access control, real-time updating, inclusivity, flexibility, and a trust anchor [31].

In such a context, in recent years, blockchain (BC) technologies have gained importance and their role in the transition towards a CE has been investigated by several researchers. According to [37], BC is an information technology that can be defined as a "digital, decentralized and distributed ledger in which transactions are logged and added in chronological order with the goal of creating permanent and tamperproof records".

Utilizing a distributed ledger architecture, BC obviates the need for a complex data center infrastructure in order to create a secure and reliable network. Such a decentralized approach facilitates the engagement of numerous stakeholders, while cryptographic algorithms are utilized to guarantee the validation and consistency of data-sharing processes [38]. Protocols for ensuring information security and facilitating updates provide the capacity to store a diverse array of data, encompassing events, records, and transactions [39]. Rusch et al. [40] highlighted how BC enables seamless information sharing, the efficient exchange of life cycle inventory data, and the real-time monitoring of product conditions. These results emphasize the potential of BC to provide transparency, traceability, and sustainability within supply chain operations, contributing to more environmentally conscious and socially responsible practices.

Consequently, BC holds the potential to facilitate a circular supply chain [41]. For example, BC can enhance the efficiency of recycling practices by enabling the effective monitoring of waste flows and by encouraging integration within recycling networks. It represents a secure and transparent platform for tracking and validating the movement of recycled materials, thereby ensuring their proper disposal or reutilization. This significantly contributes to improving the efficiency and sustainability of recycling practices and, thus, of CE [42,43].

The research conducted by Erol et al. also demonstrated that BC can play a pivotal role in promoting the adoption of a CE by enhancing supply chain traceability, fostering cooperation and coordination within business ecosystems, and fostering higher levels of trust [44]. These findings were further reinforced by Kayikci et al., who emphasized that network collaboration is the crucial factor for the successful development of a block-chain-based CE [45]. In their interviews with four small and medium enterprises (SMEs) actively involved in the field of CE, Chaudhuri et al. specifically concentrated on initiatives that employed digital technologies, such as BC, to support their transition towards a CE. The analysis revealed that SMEs participating in such initiatives were able to optimize their recycling facilities, establish clear circular economy processes, make informed decisions regarding raw materials, and optimize their manufacturing processes [46].

Moreover, BC can be used in combination with Internet of Things (IoT) by further supporting the implementation of a CE. Indeed, these technologies have the potential to enhance the efficiency of reverse supply chain systems, simplify the collection and analysis of data related to end-of-life products, and enable more accurate traceability [47]. Therefore, by leveraging these technologies, stakeholders in the circular supply chain can enhance their eco-efficiency and strengthen the resilience of the entire supply chain [48]. As an example, Magrini et al. [49] examined the application of IoT and BC technologies within the e-waste domain. Their study revealed that combining IoT and BC yields notable advantages, encompassing the capacity to track products throughout their entire lifecycle, foster CE approaches, and support well-informed decision-making processes. Indeed, organizations can effectively track usage, maintenance, and e-waste management, thereby enhancing their sustainability efforts and optimizing their resource utilization. An example of an e-waste management system based on Ethereum BC can be found in [50].

Finally, integrating BC into a supply chain offers an avenue to uphold human rights and equitable labor practices. Through transparent product history records, consumers can be confident that their purchases stem from ecologically responsible sources. Smart contracts also hold promise for autonomously enforcing the surveillance and validation of sustainable regulations and policies [51].

As a result, BC can have a pivotal role in advancing the technical aspects of the DPP.

*3.4. Types of Information to Insert in the Digital Product Passport*

Relating to the type of information to be stored in the DPP, the EC provides two categories of information:

(1)     track and trace information;
(2)     attributes.

The first category includes the set of information pertaining to the stakeholder and the associated events, such as the manufacturer's name, registered trade name, the global trade item number, TARIC code, global location number, information supporting legal compliance, etc. The second category includes the technical and sustainability related information about the product. The EC provides general examples of attributes to be included in the DPP, such as the size, color and picture of the product, origin of raw materials, environmental impact indicators, product environmental footprint, circularity indicators, social indicators, chemical content, recycled content, instructions for use, manuals, disassembly instructions, and so forth [16,17].

The following sections present how the DPP can be used as a tool for promoting human-centricity and absolute sustainability in the context of Industry 5.0.

## 4. Social Sustainability in the Digital Product Passport

The work conducted by Heikkilä and Kääriäinen highlighted the lack of social data in the DPP and the importance of their inclusion to promote social sustainability [16]. Indeed, as shown in Section 2, such an issue is overlooked in the current literature. Given the absence of an established standard methodology for conducting a social life cycle assessment of products, this study primarily relies on recommendations from the UNEP [11], and mainly refers to the PSILCA method [18], which is also endorsed by the UNEP. The rationale for this selection is that PSILCA currently stands as one of the most widely used methodologies available. As emphasized in the research available in [52], which examines the historical evolution of social life cycle assessments, PSILCA is strongly recommended for the application of the social life cycle assessment of products and for identifying social hotspots. In fact, the literature contains numerous works that rely on such a methodology. Recent examples include the studies available in [53,54]. Additionally, the 'Handbook for Product Social Impact Assessment' has also been consulted [55].

The purpose of conducting a PSILCA is to evaluate the social impacts of a product over its entire lifecycle. These impacts are evaluated in relation to different categories of stakeholders who could be either directly or indirectly affected throughout the product lifecycle. Specifically, the categories of stakeholders considered in this work are: (1) workers; (2) local community; (3) society; (4) consumers; (5) value chain actors. Each of these categories is linked to several social topics, including aspects such as health and safety, child labor, local employment, and so forth. These social topics encompass the primary social concerns of these stakeholders. The evaluation of the impact of each of these social topics is facilitated through the utilization of specific social indicators [11,18,55].

Hence, the PSILCA methodology comprises three primary steps, which are: (i) the organization of stakeholder categories; (ii) the identification of social topics for each stakeholder category; (iii) the identification of social indicators capable of evaluating how the product lifecycle affects the concerned stakeholders for that specific topic.

The following paragraphs provide examples of useful social indicators for assessing social topics for each stakeholder category interacting with a product lifecycle. Figure 2 summarizes the social indicators to be included in the DPP and it represents the answer to RQ1 (What social information will be stored in the DPP?). Each category is further described in the following sections.

| Stakeholder categories | | | | | | | | | |
|---|---|---|---|---|---|---|---|---|---|
| **Workers** | | **Local Community** | | **Society** | | **Value Chain Actors** | | **Consumers** | |
| Social topics | Indicators | Social topics | Indicators | Social topics | Indicators | Social topics | Indicators | Social topics | Indicators |
| Child Labour | Percentage of children employment. | Access to resources | Industrial water withdrawal rate. Extraction of fossil fuels, biomass, ores, and minerals per km². | Contribution to economic development | Percentage of the sector to the GDP. | Promoting Social Responsibility | Number of enterprises in initiative that promotes social responsibility. Presence of codes of conduct that protect human rights. | Health and Safety | Number of certified claims that the product contributes to a higher level of health or safety. Number of complaints identified during the reporting period related to health and safety. |
| Forced Labour | Frequency of forced labour. | | | | | | | | |
| Fair Salary | Living wage per month. | Safe and Health living conditions | CO, NMVOCs, NOx, PM10, $SO_2$, $CO_2$ emissions. | | | | | | |
| | Minimum wage per month. | | | | | | | | |
| | Sector average wage per month. | | | | | | | | |
| Working time | Working hours per day. | Local Employment | Rate of unemployment | | | | | | |
| | Average working hours per week. | | | | | | | | |
| Discrimination | Women employee rate. | | | | | | | | |
| | Difference in salary between women and men. | | | | | | | | |
| Health and Safety | Rate of fatal accidents. | | | | | | | | |
| | Rate of non-fatal accidents. | | | | | | | | |
| Social Benefits | Social security expenditure in the country. | | | | | | | Experienced well-being | Composite measure of experienced well-being. |

**Figure 2.** Social indicators to include in the DPP.

*4.1. Stakeholder Workers*

Seven main social topics concerning the workers in a product lifecycle are considered:

- Child labor;
- Forced labor;
- Fair salary;
- Working time;
- Discrimination;
- Health and Safety;
- Social benefits.

According to The World Bank [56], child labor is defined as children in employment involved in economic activity for at least one hour in the reference week of the survey. A suitable indicator to address such social issues is the percentage of child employment in the reference period. This can be further discriminated by gender via recording the percentage of male child employment and the percentage of female child employment. The data related to child employment in each relevant product lifecycle step should be recorded and stored in the DPP.

Forced Labor is defined by the International Labor Organization (ILO) as 'work or service which is exacted from any person under the menace of any penalty and for which the said person has not offered himself voluntarily' [57]. The frequency of forced labor can be used as a social indicator to represent this social topic, and it should be recorded for each relevant product lifecycle step in the DPP.

Fair wage is defined as a 'wage fairly and reasonably commensurate with the value of a particular service or class of service rendered, and, in establishing a minimum fair wage for such service or class of service.' [18]. Three indicators can provide useful information related to this social topic: the living wage per month, the minimum wage per month, and the sector average wage per month.

Working time is a social issue aiming to identify whether the number of hours that employees work is adequate both for work–life balance and for a satisfying professional life [18]. Useful social indicators for representing such information are the average working hours in a day per employee and the average working hours in a week per employee.

Worker discrimination is a very heterogenous social issue because it includes several aspects; it can be seen as inequal opportunities in education, employment, advancement, benefits, and resource distribution because of any kind of attribute unrelated to ability, performance, and qualification, such as age, race, sex, religion, political association, ethnic origin, and so forth [58]. Objectively detecting all of the aspects of discrimination using indicators is very challenging. Two examples from [18] are proposed that address gender discrimination related to employment and the gender wage gap. The former can be assessed by the ratio between the women employed in the sector and the total labor force. The latter can be evaluated as the difference in salary between women and men to assess wage disparity.

Occupational health and safety are other very important social issues to be detected and addressed in the DPP related to the worker stakeholder. The ILO defines it as 'the discipline dealing with the prevention of work-related injuries and diseases as well as the protection and promotion of the health of workers. It aims at the improvement of working conditions and environment.' [59]. Accident rates are the main indicators to represent the state of safety in the workplace conditions. They can be determined in the rate of fatal accidents, wherein death occurs within one year of the day of the accident, and the rate of non-fatal accidents, which cause injuries not causing death [18].

Social benefits basically refer to pension, disability, dependents, and survivor benefits, but can also include medical insurance, paid parental leave, education and training, etc. [18]. The social security expenditure in the country of the worker can be representative of this social issue.

*4.2. Stakeholder Local Community*

The DPP should record data and information related to social topics affecting local communities of where the product life cycle takes place. The main social topics concerning this stakeholder category are the following [11]:

- Access to material resources;
- Safe and healthy living conditions;
- Local employment.

Access to material resources aims to evaluate whether the industrial activities restrict the local community's access to resources. Indeed, according to [18], expanding operations can potentially deplete natural material resources (e.g., water, forest land, homelands) and cause conflicts over this issue, especially in emerging or unstable countries. The ratio between the industrial water withdrawal and the total water withdrawal represents the incidence of industrial water use compared to others, and it can be used as a social indicator concerning water resources. In addition to water, other material resources are also important for local communities' economy and life, such as fossil fuels, biomass, ores, and minerals. These material resources can be assessed as the total extraction of fossil fuels, biomass, ores, and minerals in tons per km$^2$ [18].

Safe and healthy living conditions for local communities are a social issue referring to the possibility of increasing the risk of disease caused by the emissions and/or poor water drainage from industrial activities. This social issue can be addressed by recording and storing environmental emissions that can cause health risks, such as Carbon monoxide (CO), Non-methane volatile organic compounds (NMVOCs), Nitrogen oxides (NOx), Atmospheric particulate matter ($PM_{10}$), Sulphur dioxide ($SO_2$), and Carbon dioxide equivalents ($CO_2$-equiv.). It is well-known that such emissions have negative impacts on the environment (acid rain, Green House Gas effect) and, in turn, cause health risks like respiratory symptoms, lung cancer, cardiovascular disease, premature delivery, birth defects, low birth weight, and premature death [18].

The local employment topic refers to the possibility of industry cooperating with the local community in order to improve their living conditions and limit the risk of both poverty and emigration. Also, cooperation with local suppliers can foster regional development. The rate of unemployment, conceived as the ratio between the number

of unemployed and the labor force, can be used as a social indicator to understand the importance of local employment in a region.

### 4.3. Stakeholder Society

Society is a broad concept that can include all of the other stakeholders. Therefore, in order to avoid overlap with the previous stakeholder categories and double counting, only the social topic of the contribution to economic development associated to the society stakeholder is inserted.

The contribution to economic development issue is related to the possibility for companies to promote economic development to society through the creation of jobs, the provision of education and training, making investments, or forwarding research [18]. A possible way to assess this social issue is through the indicator of the contribution of the sector to economic development, expressed as the percentage of a country's Gross Domestic Product (GDP). This metric reflects the creation of jobs, specific education and training, investments in businesses/infrastructure, etc. [18].

### 4.4. Stakeholder Value Chain Actors

In terms of the value chain actors, many social topics are already addressed in the worker and local community categories. According to ref. [11], this includes the social topic of promoting social responsibility.

Social responsibility refers to the broader scope of activities that a company should pursue behind the profit by addressing the interests and needs of all its stakeholders to create social value [18]. The way in which this social issue can be addressed largely varies from one organization to another. Consequently, a suitable way to measure the promotion of social responsibility along the supply chain is by the number of enterprises participating in initiatives on social responsibility, and by the existence of codes and conducts for supply chain actors concerning social responsibility [11].

### 4.5. Stakeholder Consumers

Consumers are the end-users of the product and the main social topics to be considered are the following [55]:

- Health and safety;
- Experienced well-being.

Health and safety for consumers refers to the fact that products should perform their intended functions without posing a risk to consumers' health and safety. Two indicators can be used to address this topic. The number of claims acknowledged by a certification that the product contributes to a higher level of consumer health or safety, and the number of complaints identified during the reporting period related to consumer health and safety [53].

The objective of experienced well-being is to identify the well-being that consumers experience in relation to the use of a product. The composite measure of experienced well-being serves as a suitable indicator to represent this social aspect. According to ref. [55], 'the composite measure of experienced well-being is based on experienced well-being questions. It captures aspects of the respondent's effect balance, i.e., positive and negative mood, and which of the two is the stronger. In all cases, the answers are associated with a particular experience'.

## 5. Absolute Environmental Sustainability in the Digital Product Passport

The EC already foresees some environmental sustainability attributes to be included in the DPP [17]. Environmental sustainability attributes are easier to identify compared to social data because they rely on an increasingly adopted methodology, i.e., the LCA, standardized by ISO 14044:2006. LCA analysis aims to collect the consumed resources and generated emissions throughout a product lifecycle to assess different environmental impact categories. One of the most commonly used methods to quantitively assess the envi-

ronmental impact categories of products within the LCA framework is the Environmental Footprint (EF) method, which is used as a reference in the EU [60].

However, the determination of the LCA-based environmental impact categories allows only a relative assessment between the ecological performances of technologies, without providing information on whether the technologies are sustainable in absolute terms [61]. Therefore, the identification of absolute sustainability references to be used as thresholds of environmental impacts are required. Recently, it is the application of the Planetary Boundaries (PBs) framework to LCA analysis has gained increasing attention, enabling the so-called "absolute environmental sustainability assessment" [20]. The establishment of threshold values for the environmental impact categories, represented by the PBs, would help to understand whether a product lifecycle is sustainable in absolute terms.

The current literature offers a comprehensive framework for PBs [3,62]. The study of Sala et al. [60] applied the PBs framework to the EF methodology in order to identify the allowed limits to the environmental impact indicators caused by human activities, thus enabling an absolute sustainability assessment. In particular, they defined the link between the environmental impact categories of the EF method and the PBs, as well as the numerical values of the environmental impact indicators thresholds based on the PBs, which are summarized in Table 2. Other remarkable works concerning the integration of the LCA methodology and PBs framework can be found in [2,63–65].

**Table 2.** Planetary Boundaries applied to Environmental Footprint impact categories based on [60].

| EF Impact Category | Indicator | Units of Measure | Threshold Values Based on PBs |
|---|---|---|---|
| Human toxicity, cancer | Comparative Toxic Unit for humans | CTUh | $9.62 \times 10^5$ |
| Human toxicity, non-cancer | Comparative Toxic Unit for humans | CTUh | $4.10 \times 10^6$ |
| Particulate matter | Impact on human health | Disease incidence | $5.16 \times 10^5$ |
| Photochemical ozone formation | Tropospheric ozone concentration increase | kg NMVOC eq | $4.07 \times 10^{11}$ |
| Ionizing radiation | Human exposure efficiency relative to $U^{235}$ | kBq U235 eq | $5.27 \times 10^{14}$ |
| Water use | User deprivation potential | $m^3$ world eq | $1.82 \times 10^{14}$ |
| Ecotoxicity, freshwater | Comparative Toxic Unit for ecosystems | CTUe | $1.31 \times 10^{14}$ |
| Climate change | Radiative forcing as Global Warming Potential (GWP100) | kg $CO_2$ eq | $6.81 \times 10^{12}$ |
| Resource use, fossil | Abiotic resource depletion—fossil fuels | MJ | $2.24 \times 10^{14}$ |
| Ozone depletion | Ozone Depletion Potential (ODP) | kg CFC-11 eq | $5.39 \times 10^8$ |
| Eutrophication, marine | Fraction of nutrients reaching marine end compartment (N) | kg N eq | $2.01 \times 10^{11}$ |
| Eutrophication, freshwater | Fraction of nutrients reaching freshwater end compartment (P) | kg P eq | $5.81 \times 10^9$ |
| Land use | Soil erosion | kg soil loss | $1.27 \times 10^{13}$ |
| Eutrophication, terrestrial | Accumulated Exceedance (AE) | molc N eq | $6.13 \times 10^{12}$ |
| Acidification | Accumulated Exceedance (AE) | molc Hþ eq | $1.00 \times 10^{12}$ |
| Resource use, minerals and metals | Abiotic resource depletion (ADP ultimate reserves) | kg Sb eq | $2.19 \times 10^8$ |

To make the DPP a real sustainability tool for supporting Industry 5.0, it should include, on one hand, the environmental impact indicators of each relevant product lifecycle, phase together with the associated threshold values. In this way, information relating to the sustainability of a product lifecycle can be analyzed both in absolute terms (comparing to PBs) and in relative terms (comparing to other products). On the other hand, it should also contain technical information to track the product quality throughout its lifecycle. In this way, the product information flow required to support the implementation of the CE paradigm would be fostered.

The following paragraphs provide different environmental data to be collected along a product lifecycle. Also, technical attributes to store in the DPP for each product lifecycle phase are proposed. Based on the specific product and application, some fields may be left blank.

Specifically, the following paragraphs represent the answer to the second RQ (What absolute environmental information will be stored in the DPP?).

### 5.1. Materials Sourcing Data

The sourcing process strictly adheres to the international standards concerning operational safety, environmental impact mitigation, and certification of material properties. During these stages, the information can be normalized, typically referring to kilograms (kg) or tons, and categorized as follows in the DPP [13,60,66]:

- Environmental attributes: Embodied energy, water consumption, $CO_2$ emissions, NOx emissions, Sox emissions, particulates, solid wastes, liquid wastes, etc.
- Environmental impact indicators and related PB-based thresholds: Particulate matter, climate change, resource use (fossil), water use, land use, etc., (see Table 2).
- Technical attributes: Chemical (chemical composition, toxicity, corrosion resistance, etc.), mechanical (density, elastic modulus, tensile strength, fatigue strength, etc.), thermal (thermal conductivity, thermal expansion, melting point, etc.), electrical (electrical resistivity, capacitance, dielectric constant, etc.), etc.

### 5.2. Manufacturing Data

Manufacturing operations strictly adhere to designated standards that prioritize human safety, minimize environmental impact, and ensure product performance certification. The DPP can serve as a repository for various types of information that should be recorded and stored, including the following examples:

- Environmental attributes: Processing energy, water consumption, $CO_2$ emissions, NOx emissions, Sox emissions, particulates, solid wastes, liquid wastes, etc.
- Environmental impact indicators and related PB-based thresholds: Particulate matter, climate change, resource use (fossil), water use, land use, etc., (see Table 2).
- Technical attributes: Technological (process type, etc.), geometrical (2D drawing of the part, CAD model, etc.), assembly (assembly instructions, assembly time, etc.), eventual update of previous properties.

### 5.3. Use Data

Throughout operations, products undergo changes over time due to various wear mechanisms. It is crucial to identify and monitor the most significant changes to keep the DPP up to date.

- Environmental attributes: Energy use, water consumption, $CO_2$ emissions, NOx emissions, Sox emissions, particulates, solid wastes, liquid wastes, etc.
- Environmental impact indicators and related PB-based thresholds: Particulate matter, climate change, resource use (fossil), water use, land use, etc., (see Table 2).
- Technical attributes: Operations (usage time, usage mode, product health indicators, etc.), maintenance (maintenance mode, maintenance time, etc.), update of previous properties.

### 5.4. Recovery Data

Upon reaching the end of its usage, the product's recovery activities must be monitored and incorporated into the DPP. As previously mentioned, the recovery process varies depending on the chosen strategy.

- Environmental attributes: Processing energy, water consumption, $CO_2$ emissions, NOx emissions, Sox emissions, particulates, solid wastes, liquid wastes, etc.
- Environmental impact indicators and related PB-based thresholds: Particulate matter, climate change, resource use (fossil), water use, land use, etc., (see Table 2).
- Technical attributes: Disassembly (disassembly instructions, disassembly time, etc.), update of previous properties.

### 5.5. Reusing, Reprocessing, Recycling Data

Upon the reintegration of the material/component/product into the economy, there are three potential pathways: Re-use, remanufacturing, or recycling. Depending on the

chosen scenario, relevant updates need to be made in the DPP to reflect the new information regarding usage, manufacturing, or materials sourcing.

## 6. Implementation of a DPP Platform Based on Cyber-Physical Systems

In contrast to social data, environmental and technical data can be collected objectively and automatically by exploiting the hallmark technologies of Industry 4.0, such as CPSs. The latter represent the coupling between the physical asset (product, machine, plant, etc.) and its corresponding digital world. In CPSs, data are gathered through sensors from the physical space and sent to the cyberspace, or the digital world, where data are converted into information used to guide the decision-making process [67].

According to Ahmed et al. [68], CPSs are increasingly being employed in all of the product lifecycle phases. In the material sourcing phase, most of the primary materials come from extraction processes, which are usually associated with heavy environmental impacts and risks for human health. CPSs can be used to optimally automatize the operations, which means lowering the environmental impacts and mitigating risks for humans, as well as recording the environmental and technical data associated with the process. CPSs are widely used in manufacturing to monitor machines, schedule production, and for maintenance activities. Both environmental and technical data can be recorded during such activities. Furthermore, CPSs are increasingly being used to monitor products during their usage phase to predict maintenance interventions. Even in this case, technical and environmental data can be detected and collected through sensors embedded into the products [13]. For such reasons, CPSs can be used to obtain the environmental and technical attributes along a product lifecycle and store them in the DPP.

CPSs can be conceived with several architectures, even if the so-called 5C architecture is one of the most adopted in the literature [21], and it is used as a reference for the present work. Many valuable works have already adopted the 5C architecture; some examples can be found in [69–71].

The development of the DPP will only utilize three layers. The first layer is called the Connection Level and it is represented by the asset to be monitored together with the sensors used to gather the data. The collected data are sent to the second layer, constituted by the Conversion Level, where the data are transformed into information through the utilization of statistical analysis and machine learning algorithms. The Cyber Level represents the third layer, where the data and information are used to create knowledge, mainly through data analytics and artificial intelligence tools.

In this context, each stakeholder can use the Connection Level of its CPS to gather the technical and environmental attributes needed in the DPP. The Conversion Level is responsible for processing such data and forwarding them to the Cyber Level, where technical parameters, as well as environmental impact indicators based on the EF methodology and the related thresholds, will be made available. Thus, each stakeholder's cyberspace will contain the relevant product technical parameters and environmental impact indicators with the associated PBs thresholds, to be sent to the DPP. This can be conducted, for example, through means of decentralized technologies, as explained in the Section 3 of this article. Figure 3 displays the conceived framework for the digitalization of the product value chain and for filling in the DPP considering a generic product lifecycle phase as an example. It represents part of the answer to the third RQ of this work (How can the product lifecycle be digitalized to collect useful data for the DPP?).

The classical model for circular value chains is adopted, which is composed of the following four phases: Materials Sourcing, Manufacturing, Use, and Recovery. The stakeholders involved in materials sourcing establish a connection between their cyberspace and the DPP to transmit the identified environmental and technical attributes. When the materials flow to the next stakeholders, belonging to the manufacturing phase, the stakeholders will update the DPP by adding new environmental and technical attributes from their cyberspace.

Upon delivery of the product to the end-users, information can be collected through various means. If the product is equipped with sensors, the DPP can be automatically updated by gathering the environmental and technical attributes through sensor data. Otherwise, the technical attributes can be directly detected at the end of the usage phase using Non-Destructive Testing (NDT) techniques, including computed tomography, ultrasonic testing, thermographic inspection, and other applicable methods. Furthermore, reverse engineering techniques, such as 3D scanning systems, can be employed to identify changes in the product geometry resulting from deterioration mechanisms. These approaches contribute to updating the DPP with accurate and relevant information.

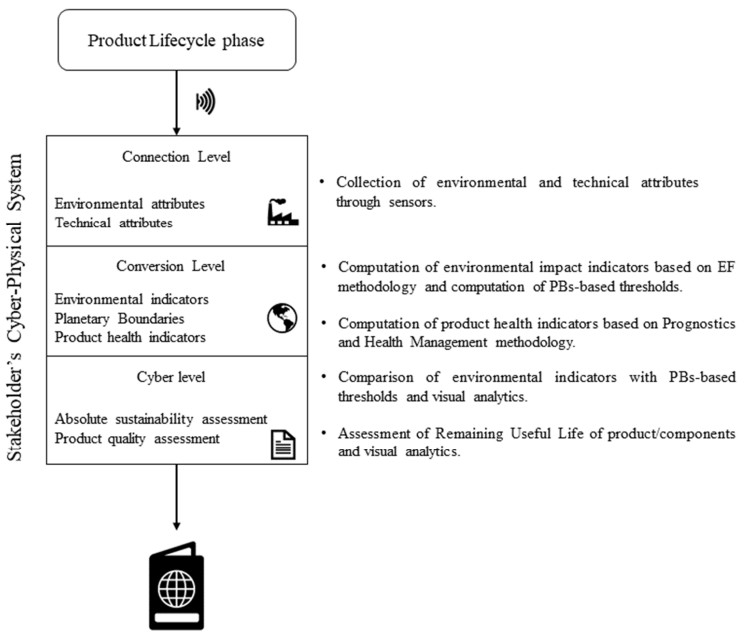

**Figure 3.** Use of stakeholder's Cyber-Physical System to fill the DPP.

At the end of the Use phase, according to the selected Recovery scenario—which can roughly be summarized as reusing, reprocessing, or recycling—new environmental and technical attributes can be recorded and the stakeholders in charge of that activity will update the DPP accordingly.

Figure 4 sketches the conceived framework applied to the entire circular value chain. It complements the answer to the third RQ of this work (How can the product lifecycle be digitalized to collect useful data for the DPP?).

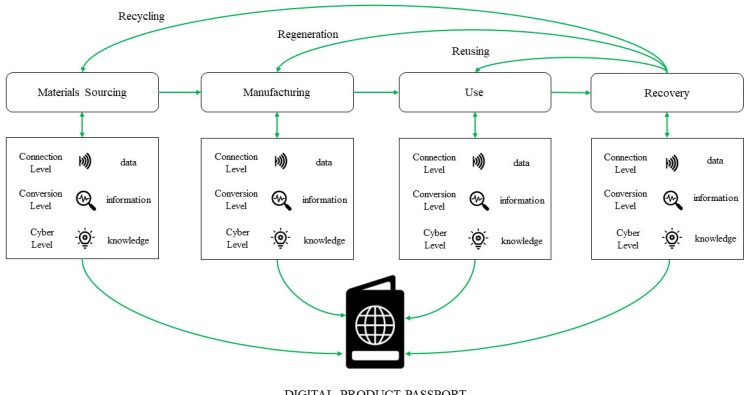

**Figure 4.** Use of CPSs for filling the DPP throughout the product lifecycle.

## 7. Conclusions

In this work, the current state of the DPP has been presented and the need for integrating social sustainability and absolute environmental sustainability into it has been presented. Specifically, following the UNEP's recommendation and utilizing the PSILCA methodology, a collection of social indicators to include in the DPP has been proposed to facilitate a social assessment of a product's lifecycle. Subsequently, a set of environmental attributes and environmental impact indicators has been proposed, employing the LCA methodology, along with the corresponding threshold values based on the PBs framework. These proposed additions to the DPP will allow for the assessment of products' absolute environmental sustainability.

These concepts, which are here introduced for the first time in the DPP literature, aim to foster a more holistic decision-making process for stakeholders in a product value chain. Indeed, the present work aims to complement the current frameworks and methodologies available in the literature regarding the DPP, rather than replace them. Furthermore, a framework based on cyber-physical systems to digitalize the product passport has been proposed, enabling the collection of environmental and technical information throughout the product lifecycle. This digitalization process allows stakeholders in the product value chain to store and share all of the necessary product-related information required for assessing its absolute environmental sustainability.

Moving forward, several future research directions are outlined. Firstly, the design and implementation of a software platform to support digital product passports will be crucial, considering aspects such as data security and standardization. To address data security concerns and protect the stakeholders engaged in data sharing, the adoption of crypto-anchors and BC technologies can be considered, as they provide robust mechanisms for safeguarding data. Additionally, seamless data integration across various value chains should be ensured to maintain coherence and consistency.

Another research area of importance pertains to the collection of social data within the DPP. As social data are subjective and influenced by various cultural values and perceptions, standardization is necessary to determine the responsible party for collecting and documenting such data in the passport. However, according to the UNEP [11], the collection of social data typically relies on licensed databases like PSILCA. Alternatively, when primary data are required, it can be obtained by visiting specific production sites or through collaboration with the relevant organizations. Primary data can be gathered through direct interaction with organizations and companies, by working with NGOs or similar organizations, by observing on-site business/production processes, or by conducting interviews or surveys with the affected stakeholders (e.g., workers).

Lastly, it is vital to assess the sustainability of the digitalization process itself. While digitalization offers valuable opportunities for a CE, it is important to recognize that Information and Communication Technology (ICT) relies on a significant amount of energy, which contributes to environmental emissions. Thus, efforts towards digitalization must be accompanied by a simultaneous increase in the utilization of renewable energy sources and improved energy efficiency. This coupling is essential to mitigate the environmental impact associated with ICT technologies.

**Author Contributions:** Conceptualization, L.P., G.B. and F.L.; methodology, L.P., G.B. and F.L.; data curation, L.P., G.B. and F.L.; writing—original draft preparation, L.P. and G.B.; writing—review and editing, L.P., G.B. and F.L. All authors have read and agreed to the published version of the manuscript.

**Funding:** This research received no external funding.

**Institutional Review Board Statement:** Not applicable.

**Informed Consent Statement:** Not applicable.

**Data Availability Statement:** The data presented in this study are available in the manuscript text.

**Conflicts of Interest:** The authors declare no conflict of interest.

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
