# Peer review of "Integrating Absolute Sustainability and Social Sustainability in the Digital Product Passport to Promote Industry 5.0"

_sustainability, doi:10.3390/su151612552_

Round 1

Reviewer 1 Report

The paper “Integrating absolute sustainability and social sustainability in the digital product passport to promote industry 5.0” investigates useful social indicators to include in the digital product passport for 12 enabling a social assessment of a product lifecycle. Besides, the authors propose a framework based on the cyber-physical system for filling 17 in the digital product passport throughout a product lifecycle. To achieve this aim, no methodology is presented; this is a point to add to the abstract to inform the readers what they must expect in the rest of the manuscript.

I find the chosen topic very interesting and contemporary, but the manuscript needs further efforts to be improved and ready for publication.

The introduction deals with Sustainable Development Goals and then moves on to Industry 4.0 all too suddenly. Moving from one topic to another more linearly and logically would be necessary.

Furthermore, it is said that “This can be achieved by addressing the two main drivers of environmental impacts, i.e., the energy consumption and the resources consumption.” This sentence has a reference or is an assumption of the author(s); in this case, it is needed to be better explained.

As for the theoretical background, I suggest clarifying how the literature on the topic was reviewed to justify the research framework according to relevant theories in the body of literature. The authors should improve the theoretical justification of the study. Which is/are the theory/ies adopted to justify the study? Moreover, the authors present an introduction section in which the integration of social, managerial and environmental issues should be better analysed. In fact, in this present version, this section neglects some valuable recent contributions in the body of knowledge (particularly on blockchain aspects and social aspects of SDGs), and therefore the authors should work more to update the literature background. I advise authors also to improve the theoretical discussion with more recent and relevant references.

Section 3 deals with digital decentralised identifiers, decentralised storage, verifiability of the data, and many other features related to blockchain technology, reducing the discussion of this technology to a few lines. It would be necessary to describe better how such technology could support the objectives related to the digital product passport in trust, transparency, traceability and disintermediation.

Section 4 talks about the centrality of man in the digital product passport, but it looks a bit general. The variables to consider come from the various standards divided into macro-categories provided by the Guidelines for Social Life Cycle Assessment of Products and Organizations (2020). For the sake of clarity of the text, it is necessary to describe the second-level variables further, better explaining their size, meaning, how to measure them, and from which standard such variable is taken.

The methodology is not described in the manuscript for the review or the proposed conceptual model. This is the weakest point in this work.

The conclusions are very short. It would be necessary to better describe the results obtained, both from the taxonomy side and from the side of the proposed framework. Furthermore, a distinction between theoretical and managerial contributions would also be appropriate to distinguish how the work contributes to the two aspects.

As for the future contributions and implications, I also suggest highlighting the potential role of sustainability and digital transition issues as a core direction for the topic investigated. I suggest to use also them as the perspective of analysis in discussing future research directions.

Concerning the context of an investigation, could it affect the results? Why? Additional information on generating the evaluation should be included to justify the results' significance and clarify the proposed approach's effectiveness and rationality. Moreover, once they stated what methodology they are using, the author/s should describe more in detail if and how their application converges or diverges from other approaches. The authors should also improve the theoretical justification according to previous methodological contributions.

Finally, proofreading in different sections is needed.

Good luck.

Proofreading in different sections is needed.

Author Response

We are grateful to the Reviewer for his/her valuable suggestions aimed to improve the quality of our work.   In the following we provide our point-by-point reply:   - the abstract has been modified to better emphasize the contribute of the manuscript in respect to social indicators, absolute environmental indicators, and the proposed framework for product lifecycle digitalization based on CPSs. - the introduction has been reduced and some parts have been removed according to the requests of another reviewer. - the literature was analyzed by consulting the articles available in the Google Scholar database concerning the keyword 'Digital Product Passport' and by referring only to journal articles and conference papers, in English language. The papers have been read, and the most relevant have been included in the literature section of the present work. Such explanation has been added in the article. Furthermore, a table (Table 1) has been added at the end of the Section 2 to better illustrate the current works available in the literature. - Section 3.3 has been more detailed as requested by the reviewer with a paragraph supported by the literature relating to the relevance of blockchain technology for circular supply chains. - each social indicator proposed in section 4 has a definition supported by the literature - the objective of the work is threefold: (i) the inclusion of social indicators in the DPP; we address this point by referring to three main social life cycle assessment methodologies; (ii) the inclusion of absolute environmental sustainability in the DPP; we address this point by referring to the Environmental Footprint methodology and the planetary boundaries framework; (iii) the development of a conceptual framework for the digitalization of the product lifecyle that is conceived by the authors by referring to a well-known CPS architecture, 5C, as reported in the main text. - the conclusions have been extended and rearranged as also requested by other reviewers. - our contribution doesn't replace the previous ones, but it complements them. So, the present article can be a valuable contribution to enrich the other methodologies available in the literature by including social sustainability and absolute and environmental sustainability.

Reviewer 2 Report

Thanks for assigning me this manuscript for revision. Although the manuscript is written in good, a few points need further clarification.

Abstract: Please also try to avoid the I & We in this and the remaining sections.

Introduction: The section is too long. The point that needs to be focused on is study motivation, what is missing in the past literature (gaps), why this study is so important, and what are the novel contributions.

Table 1, the unit and PB column, as a new researcher, can they understand what is meant by unit and PB. Please clarify this further.

In the Material sourcing data section, please include the source and references.

Discussion and conclusion:  Please justify your findings and support from the literature. What are the novel findings and why the findings are different from others? 

Thanks for assigning me this manuscript for revision. Although the manuscript is written in good, a few points need further clarification.

Abstract: Please also try to avoid the I & We in this and the remaining sections.

Introduction: The section is too long. The point that needs to be focused on is study motivation, what is missing in the past literature (gaps), why this study is so important, and what are the novel contributions.

Table 1, the unit and PB column, as a new researcher, can they understand what is meant by unit and PB. Please clarify this further.

In the Material sourcing data section, please include the source and references.

Discussion and conclusion:  Please justify your findings and support from the literature. What are the novel findings and why the findings are different from others? 

Author Response

We thank the Reviewer for his/her valuable suggestions aimed to improve the quality of our work.   In the following we provide our point-by-point reply:   - we have removed the words 'I' and 'we' in the abstract as requested by the reviewer. The abstract has been furtherly rearranged as requested by other reviewers. - we have shortened the introduction by keeping the focus on the study motivations, the gaps, and the contribution of the work; - we have better clarified the last 2 table column headers; - we have added the references in materials sourcing section as required by the reviewer; - we have modified the conclusion section to better emphasize our contributes and future research directions.

Reviewer 3 Report

Very interesting paper. The authors propose a set of social indicators to include in the DPP. The title of the paper refers to the integrating social sustainability, but this is not explicitly stated in the text. Section 6 describes the framework for the automatic collection of environmental and technical data throughout the life cycle of a product and does not include social data..

On the line 642 is written "Another research area is related with the integration of the social data in the digital product passport. ", ……… so the title of the paper needs to be edited.

Line 323: poorly readable; the identification is not correct, this is a table and this is not a figure

Line 597 and 619: incorrect figure numbering

Author Response

We thank the Reviewer for his/her valuable suggestions aimed to improve the quality of our work.   In the following we provide our point-by-point reply:   - in this work, we have proposed some useful social indicators to include in the DPP. They are defined and discussed in a proper section, and they are supported by the literature relating to the social life cycle assessment. Because of this, the authors think that it's not necessary to change the title. However, we have rearranged the sentence in the conclusion section to better clarify such point. - we have renumbered the tables and figures according to the reviewer requests.

Reviewer 4 Report

Dear authors,

Your research on DPP to promote Industry 5.0 is a very interesting and topical subject. Below you will find some suggestions to improve your article:

You mention the term "absolute sustainability" and "absolute environmental sustainability" interchangeably. Is this term yours or has it been presented and conceptualised by more authors? For me it is confusing because there is another term, called Tripple Bottom Line (TBL), which includes economic, social and environmental issues. Therefore, for me, "absolute sustainability" should pay attention to these 3 issues, not only to environmental sustainability. So please consider adding the term TBL and redefine "absolute sustainability" or better explain the term absolute sustainability in the introduction (line 66) using references to this concept.

Also, in lines 128-137 if you introduce the TBL concept, you could highlight the importance of measuring the social and environmental dimensions of sustainability.

Finally, I miss a use case on the application of the DPP in an SPC.  Without this use case example, the article is a review of DPP and CPS without following any review research methodology, so in my opinion it impoverishes the research. If you don't have any SCP implementation use cases yet, it would be interesting to present a review of cases/research by other authors.

Apart from that, check the Figures because Figure 1 appears twice.

Author Response

We thank the Reviewer for his/her valuable suggestions aimed to improve the quality of our work.   In the following we provide our point-by-point reply:   - in the introduction section we have better clarified the definition of absolute environmental sustainability by adopting references from the literature.

- the goal of the article is not to propose a literature review, because of this we didn't follow a specific methodology in revising the literature. Instead, the goal of our article is threefold: (i) to propose social indicators to include in the DPP, (ii) to propose environmental indicators based on Environmental Footprint methodology and the planetary boundaries framework, and (iii) to propose a framework for product life cycle digitalization. We have clarified such point directly in the Abstract. However, as requested by the reviewer, we have better specified the contributions available in the literature by inserting a table (Table 1) at the end of the section 2.

Round 2

Reviewer 1 Report

I appreciate the authors' efforts to enhance the quality of the presented research. In some parts, clarity and scientific rigor are enhanced. Unfortunately, the methodological issue persists. As in the initial draught, it remains impossible to comprehend the purpose of the work. Although you explain it clearly in the introduction, the following paragraphs leave confused. In fact, it is almost as if your research culminates in a taxonomy for which you propose a verification framework. In this situation, it would be necessary to structure a review methodology in this manner. In this case, there is a complete absence of a reference theory in the field of entrepreneurial management or technological innovation that theoretically supports you, as well as qualitative research that supports and/or enhances the proposed framework.

This fundamental weakness implies that the conclusions are neither relevant and effective (for an empirical work) nor scientifically supported (given the lack of a strict protocol for literature review).

Currently, the work does not appear to be ready for publication.

Hoping you will take my doubts positively, I wish you good luck.

Minor editing of English language required

Reviewer 4 Report

Dear Authors,

I have no more suggestions.

Best Regards

Author Response

Dear Review, 

Thank you for the appreciation.

Best regards,

The Authors

Round 3

Reviewer 1 Report

The paper “Integrating absolute sustainability and social sustainability in the digital product passport to promote industry 5.0” improved after the last revisions, even though a few further adjustments look necessary.

It looks necessary to better explain the Product Social Impact Life Cycle Assessment methodology from a theoretical point of view since it is a new methodology.

Table 2 looks sparse or incomplete; please check if it is well filled.

I also advise improving the theoretical discussion with more recent and relevant references (it looks like you often used the same references to justify your assumptions). 

Section 3 deals with digital decentralised identifiers, decentralised storage, verifiability of the data, and many other features related to blockchain technology. I agree with the relevance of blockchain technology for digital passports, but it is necessary to clarify better how blockchain is a pivotal technology in supply chain systems to support trust, transparency, and traceability, particularly from a circular economy perspective. 

Good luck.
